# Adaptive Image Deblurring Convolutional Neural Network with Meta-Tuning

**DOI:** 10.3390/s25165211

**Published:** 2025-08-21

**Authors:** Quoc-Thien Ho, Minh-Thien Duong, Seongsoo Lee, Min-Cheol Hong

**Affiliations:** 1Department of Information and Telecommunication Engineering, Soongsil University, Seoul 06978, Republic of Korea; hoquocthiendl@soongsil.ac.kr; 2Department of Automatic Control, Ho Chi Minh City University of Technology and Education, Ho Chi Minh City 70000, Vietnam; minhthien@hcmute.edu.vn; 3Department of Intelligent Semiconductor, Soongsil University, Seoul 06978, Republic of Korea; sslee@ssu.ac.kr; 4School of Electronic Engineering, Soongsil University, Seoul 06978, Republic of Korea

**Keywords:** blur domain adaptation, convolution neural networks, deep learning, image deblurring, imaging sensor, motion blur, small kernel size, receptive field, undesired artifacts

## Abstract

Motion blur is a complex phenomenon caused by the relative movement between an observed object and an imaging sensor during the exposure time, resulting in degradation in the image quality. Deep-learning-based methods, particularly convolutional neural networks (CNNs), have shown promise in motion deblurring. However, the small kernel sizes of CNNs limit their ability to achieve optimal performance. Moreover, supervised deep-learning-based deblurring methods often exhibit overfitting in their training datasets. Models trained on widely used synthetic blur datasets frequently fail to generalize in other blur domains in real-world scenarios and often produce undesired artifacts. To address these challenges, we propose the Spatial Feature Selection Network (SFSNet), which incorporates a Regional Feature Extractor (RFE) module to expand the receptive field and effectively select critical spatial features in order to improve the deblurring performance. In addition, we present the BlurMix dataset, which includes diverse blur types, as well as a meta-tuning strategy for effective blur domain adaptation. Our method enables the network to rapidly adapt to novel blur distributions with minimal additional training, and thereby improve generalization. The experimental results show that the meta-tuning variant of the SFSNet eliminates unwanted artifacts and significantly improves the deblurring performance across various blur domains.

## 1. Introduction

The relative movement between the camera and objects during image capture generates motion blur artifacts, degrading image aesthetics and high-level computer vision performance, such as emotional recognition [1], medical imaging [2,3], and autonomous driving [4]. In these applications, image clarity is crucial, and a slight blur may ruin essential information, leading to serious consequences. In addition, because motion blur is usually nonuniform and unpredictable [5], it is difficult to accurately model blur patterns.

Traditionally, non-deep learning (non-DL) approaches address image deblurring by modeling a blurred image as a convolution of a sharp image with a blur kernel and then solving it as an inverse filtering problem [6,7,8,9,10,11,12,13,14]. However, these methods typically rely on specific assumptions regarding motion blur [15,16], which constrain their effectiveness when dealing with complex blur patterns and dynamic scenes frequently observed in real-world scenarios.

Recent advances in graphics processing units (GPUs) and large-scale datasets have driven significant progress in low-level vision tasks through deep-learning-based models, particularly convolutional neural networks (CNNs) [17,18,19]. CNN-based image deblurring models trained on large-scale datasets [20,21,22,23,24,25,26,27,28,29,30] have demonstrated exceptional performance. However, conventional CNNs are inherently limited by two primary factors. First, their fixed weight-sharing scheme imposes spatial inflexibility during feature extraction [31]. Second, the small receptive fields of standard 3×3 kernels prevent the effective capture of long-range feature dependencies [32]. Dilated convolutions [33] were introduced to address the constraints of small kernel sizes and weight sharing in CNN-based deblurring by enlarging the receptive field, yielding promising results [27,34,35]. Nevertheless, dilated convolutions often cause checkerboard artifacts, which compromise the fine-grained details. In addition, blurry images can originate from various sources, including kernel convolution, frame averaging, and direct capture from real-world environments. Consequently, deep-learning-based (DL-based) models trained on specific synthetic datasets such as the widely used GoPro dataset [20] often struggle to generalize to other blur domains, particularly those encountered in real-world scenarios. This shortfall leads to unwanted artifacts in the restored images or subpar deblurring performance, as illustrated in Figure 1. A straightforward solution to mitigate this issue is to retrain the models on new datasets. However, this method is computationally expensive, time-consuming, and frequently ineffective for unseen blur patterns. Recent studies [36] have incorporated blur kernel estimation into DL-based deblurring models to improve the robustness.

Overall, previous CNN-based image deblurring models suffer from two primary issues: (1) limited receptive field and spatial inflexibility, and (2) poor generalizability in real-world scenarios when trained solely on synthesized datasets, such as GoPro.

To address these issues, we propose the Spatial Feature Selection Network (SFSNet), a CNN-based model optimized for effective and efficient image deblurring. SFSNet integrates a Local Feature Extraction (LFE) module to capture fine-grained local details and a Regional Feature Extractor (RFE) module to capture broader contextual information. Specifically, the RFE leverages 5×5 and 7×7 depthwise convolutions followed by a gated convolution to expand the receptive field and dynamically emphasize spatially relevant features, thereby substantially improving deblurring performance. To overcome the limited diversity of the GoPro dataset, we introduce BlurMix, a curated dataset comprising blur–sharp image pairs drawn from multiple deblurring datasets, offering a broader spectrum of real-world blur characteristics. Moreover, we propose a meta-tuning strategy to tune the pretrained SFSNet on the additional BlurMix dataset. This approach enables SFSNet to effectively adapt to various blur domains within a single epoch, reduces artifacts, and further enhances deblurring robustness.

In summary, the key contributions of this study are as follows:The SFSNet proposed in this study integrates an RFE to address the limitation of the traditional 3×3 convolutions by expanding the receptive field and adaptively selecting spatial features to improve the deblurring performance.The introduction of the BlurMix dataset mitigates the limited generalizability of existing datasets, accompanied by a meta-tuning strategy that enables rapid adaptation to various blur domains.The experimental results demonstrate that DL-based models, such as the SFSNet with meta-tuning, effectively reduce undesired artifacts and enhance the deblurring performance across diverse datasets.

The remainder of this paper is organized as follows: The DL-based deblurring methods and relevant datasets are reviewed in Section 2. Our proposed method is described in Section 3. The experimental results and analyses are presented in Section 4. Finally, the conclusions of this study are presented in Section 5.

## 2. Related Work

Deep-learning-based models are highly dependent on the quality and diversity of their training datasets. This section provides an overview of the existing deep learning architectures for image deblurring and explores techniques for constructing relevant datasets.

### 2.1. Deep-Learning-Based Image Deblurring

Early applications of deep learning for image deblurring use CNNs to estimate blur kernels and apply deconvolution for image restoration [37,38]. However, these methods rely on simplified blur assumptions and struggle to capture the complex blur patterns found in dynamic real-world scenes. The limitations of kernel-based CNNs have prompted the development of more advanced approaches. Nah et al. [20] introduced a pioneering end-to-end CNN framework trained on the GoPro dataset, which is a large-scale dataset containing blur-sharp image pairs for image deblurring. This study inspired subsequent research on multiscale architectures aimed at improving both the performance and efficiency of image deblurring [20,21,22,23,24,25,26,27,28,29,30]. Despite significant improvements, conventional CNN-based models typically use a fixed 3×3 kernel size, limiting their receptive fields. Furthermore, the weight-sharing properties of CNNs restrict their ability to adapt to spatially varying blur patterns. To address these challenges, dilated convolution was introduced for image deblurring. Zou et al. [35] employed dilated convolutions with various dilation rates and wavelet reconstruction modules to expand the receptive field and recover high-frequency details. Tsai et al. [27] explored region-based self-attention alongside parallel dilated convolution to disentangle blurred patterns of different magnitudes and orientations while aggregating multiscale information. Although dilated convolution enlarges the receptive field, it can introduce checkerboard artifacts, resulting in the loss of fine-grained local details. Furthermore, the inherent weight-sharing properties of CNNs [31] limit spatial adaptability, making it challenging to handle highly dynamic blur variations in real-world images. In addition, vision transformers [39] have recently emerged as a powerful approach to capture global information in images, achieving significant success in various computer vision and image processing tasks. In image restoration, transformer-based models utilize self-attention mechanisms, which enable efficient modeling of long-range dependencies and yield notable performance improvements [32,40]. However, these models come with substantial computational and memory costs that scale with the input size. To address this challenge, Liu et al. [41] drew inspiration from vision transformers to enhance convolutional neural network (CNN) architectures by incorporating large kernel sizes. This approach achieves a compelling balance of efficiency and performance in image classification, paving the way for modernized CNN-based models that retain transformer-inspired insights while maintaining computational efficiency.

Moreover, the widely used GoPro dataset [20], synthesized using frame averaging, often produces unrealistic and discontinuous blur patterns. Consequently, data-driven deblurring models trained exclusively on this dataset often struggle to handle blurry images in real-world scenarios [42]. Several methods have been proposed to improve the generalizability of the models. The UFPNet [36] integrates blur-kernel estimation branches into an end-to-end deep CNN to enhance the robustness of models trained on the GoPro dataset. In this study, we propose a meta-tuning strategy to adapt deep-learning-based image deblurring models to diverse blur domains, thereby enhancing the robustness and deblurring performance across diverse datasets.

### 2.2. Deblurring Datasets

Early deblurring methods treated blurry images as degradation process models using kernel convolution [6,7,8,9,10,11,12,13,14]. This approach represents a blurry image as(1)B=k∗S+n,
where B,k,S, and *n* represent the blurry image, blur kernel, sharp image, and additive noise, respectively. However, these methods typically assume a spatially uniform blur kernel and limited scene complexity and do not capture the inherently nonuniform, dynamic nature of blur in real-world scenarios. The rapid development of CNNs for image processing has encountered a critical obstacle, which is the scarcity of large-scale datasets specifically designed for image deblurring. Traditional deblurring methods rely on limited datasets, which are insufficient for data-intensive deep-learning approaches. Recognizing this need, the GoPro dataset [20] emerged as a practical solution. The GoPro dataset redefines blur modeling by approximating motion blur through frame averaging, thereby simplifying the complex mathematical representation of blur as the temporal integration of sharp images over time, expressed as(2)$$B=g\left(\frac{1}{T}\int_{t=0}^{T}S\left(t\right)dt\right)≃g\left(\frac{1}{N}\sum_{i=0}^{N-1}S\left[i\right]\right)\;,$$
where *B* denotes the blurry image, S(t) represents the sensor signal of the sharp image at time *t*, *T* indicates the exposure time, and g(.) is the camera response function. In addition, *N* denotes the number of frames sampled, and S[i] represents the *i*-th sharp frame. This approach encourages the generation of large-scale deblurring datasets [43,44], paving the way for powerful deep-learning-based deblurring models [20,21,22,23,24,25,26,27,28,29,30]. However, recent research [42] demonstrated that datasets acquired using the frame averaging approach may result in discontinuous motion and unrealistic blurring. Consequently, models trained solely on such datasets may struggle to generalize the complexities of real-world blurry images. To address this issue, the previous studies [42,45,46] introduced datasets that captured real-world image pairs using elaborate hardware setups involving two parallel cameras with different shutter speeds. In addition, the RWBI real-world blurry dataset [22] was introduced for evaluating deblurring models. In summary, image deblurring datasets can be acquired using various methods, each with distinct blur properties and acquisition techniques. Table 1 presents an overview of existing blur datasets and their characteristics.

In this study, we present the BlurMix dataset, which is constructed by randomly sampling blur-sharp image pairs from multiple blur domains to capture a diverse range of blur patterns. By applying meta-tuning, our approach effectively mitigates undesired artifacts and improves deblurring performance across various blur domains.

## 3. Proposed Method

In this section, we present the detailed structure of the SFSNet, the BlurMix dataset, and the meta-tuning strategy for blur domain adaptation.

### 3.1. Spatial Feature Selection Network (SFSNet) for Image Deblurring

Conventional CNNs have limitations owing to their restricted receptive fields and rigidity imposed by fixed 3×3 convolutional kernels. To overcome these drawbacks, we introduced the Spatial Feature Selection Network (SFSNet), which is a deblurring model developed to enhance both feature extraction and spatial adaptability. The SFSNet incorporates local feature extractors (LFE) to capture local features and regional feature extractors (RFE) to expand the receptive field and adaptively select critical spatial features. This design effectively addresses the challenges associated with conventional CNN-based deblurring models. The architecture of the SFSNet follows a multi-scale encoder-decoder structure, incorporating LFE and RFE at various levels to improve the deblurring performance, as illustrated in Figure 2.

The deblurring process of the SFSNet can be described as(3)$$\hat{I}=f\left(B\right)=R+B\;,$$
where $\hat{I}$ represents the restored images, f(·) is the SFSNet deblurring function, and *R* and *B* denote the residual and input blurry images, respectively. The LFE uses a standard 3×3 kernel size for local feature extraction. The RFE integrates a Spatial Gated Module (SGM) that employs two depthwise convolutional branches with 5×5 and 7×7 kernels to expand the receptive field and capture richer spatial information. This selection in the SGM balances the capture of local and mid-range spatial dependencies while maintaining computational efficiency. The extracted features from the two spatial feature maps are then refined through gated convolution [47], which selects the most relevant regional features for further processing:(4)$$SGM\left(X\right)=W_{d}^{5\times 5}\left(X\right)⊗\sigma \left(W_{d}^{7\times 7}\left(X\right)\right)\;,$$
where $W_{d}^{5\times 5}\left(\cdot \right)$ and $W_{d}^{7\times 7}\left(\cdot \right)$ are depthwise convolutions with kernel sizes of 5×5 and 7×7, respectively, ⊗ is the element-wise product and *X* is the input feature map. In addition, σ represents the activation function, which is empirically set as the identity function in this case. The training process of the SFSNet is conducted through supervised learning using blurry and sharp image pairs. The overall loss function of SFSNet is formulated as(5)$$L_{SFSNet}=L_{1}+\lambda L_{frequency}\;,$$
where $L_{1}$ represents the pixel-wise reconstruction loss, $L_{frequency}$ captures the frequency-domain discrepancies, and λ is a weighting parameter empirically set to 0.1. $L_{1}$ is applied to align the primary structure of the restored image $\hat{I}$ with the ground truth *S*, which is given by(6)$$L_{1}={\left|\left|\hat{I}-S\right|\right|}_{1}\;.$$

In addition, $L_{frequency}$ [24] is used to capture and preserve the frequency information, which is described as(7)$$L_{frequency}={\left|\left|F\left(\hat{I}\right)-F\left(S\right)\right|\right|}_{1}\;,$$
where F(·) is the fast Fourier transform function.

### 3.2. Generation of the BlurMix Dataset

#### 3.2.1. Existing Image Deblurring Datasets

Deep-learning-based models are typically trained on a specific large-scale synthetic dataset, such as the GoPro dataset [20], which is a widely used dataset for image deblurring and excellently restores blurry images in its blur domain. However, they often struggle to deblur images with unknown blur characteristics in real-world scenarios [42]. In particular, deep-learning models such as MIMOUNet+ [24], BANet+ [27], and FSNet [29] exhibit poor deblurring performance when trained on the GoPro dataset [20] and testing on real-world datasets, including RealBlur [42] and RSBlur [45], as demonstrated in Table 2.

In this study, we categorized existing deblurring datasets based on their acquisition methods and selected nonuniform datasets that are suitable for training and testing deep-learning-based models for image deblurring, as shown in Table 3.

Blur kernel convolution: MC-UHDM [14] is a large-scale high-resolution dataset acquired by convolving sharp images with large blur kernel sizes ranging from 15×15 to 191×191.Frame averaging: The GoPro [20], HIDE [43], and REDS [44] generate blurry images by averaging successive sharp frames. A ground truth image is selected from the center of these frames. Notably, the REDS were selected with different background environments and camera settings compared with those of the GoPro and HIDE datasets.Real-world shooting: The RealBlur [42], RSBlur [45], and ReLoBlur [46] are collected using a system of cameras that simultaneously capture pairs of blurry and sharp images with geometrical alignment. The RealBlur dataset [42], comprising RealBlur and RealBlur-Tele subsets, was obtained using wide-angle telephoto lenses, which produce severe blurriness. The RSBlur dataset [45] is a real-world dataset that provides large-scale pairs of blurry and sharp high-resolution images, whereas the ReLoBlur dataset [46] is the first real-world local motion deblurring dataset that includes indoor and outdoor scenes of diverse motion objects. The RWBI dataset [22] was collected using handheld devices and used for qualitative evaluation.

#### 3.2.2. BlurMix Dataset

Subsequently, we introduce the BlurMix dataset, which is a comprehensive collection curated from existing blur datasets to encompass both synthetic and real-world scenarios. The BlurMix dataset gathers image pairs from various widely recognized deblurring benchmarks, each of which illustrates a unique blur generation method. Frame averaging in REDS [44] datasets captures motion-induced blur in dynamic scenes, whereas kernel convolution in the MC-UHDM dataset [46] provides high-resolution synthetic blur. The RealBlur [42], RealBlur-Tele [42], and RSBlur [45] datasets offer genuine camera-based blurring. By integrating these diverse sources, the BlurMix dataset provides a balanced dataset that supports the robust advancement of blur domain adaptation methods. Table 4 lists the number of image pairs, resolutions, and acquisition methods.

### 3.3. Meta-Tuning Strategy for Adaptive Image Deblurring

In this section, we introduce a meta-tuning strategy designed to improve the generalization capabilities of a pretrained image deblurring model. This strategy leverages a meta-learning approach [48] to enhance the adaptability of the model across diverse motion blur patterns by exploiting two complementary datasets: GoPro and BlurMix. Algorithm 1 outlines the meta-tuning process for adaptive image deblurring.
**Algorithm 1** Meta-Tuning for adaptive image deblurring**Require:** Training datasets of GoPro and BlurMix ($D_{\mathrm{GoPro}}$ and $D_{\mathrm{BlurMix}}$) where *x* and *y* are blur and sharp images, respectively; step size hyperparameters for inner and outer loops (α and β ); number of inner steps (*K*)Initialize pretrained SFSNet $f_{\theta }$ on the GoPro datasetSample batches $B_{i}^{\mathrm{GoPro}}∼D_{\mathrm{GoPro}}$ with batch size $N_{GoPro}$Sample batches $B_{k}^{\mathrm{BlurMix}}∼D_{\mathrm{BlurMix}}$ with batch size $N_{BlurMix}$**(Outer loop-Deblurring Learning on GoPro)****for** all $B_{i}^{\mathrm{GoPro}}$ **do**    Clone parameters: $\theta^{\prime }\leftarrow \theta$    **(Inner loop-Deblurring Domains Adaptation with BlurMix)**    **for** K inner steps **do**        Evaluate $\nabla_{\theta }L_{\mathrm{inner}}\left(f_{\theta }^{\prime }\right)$ with $L_{\mathrm{inner}}=\frac{1}{N_{BlurMix}}\sum_{\left(x,y\right)\in B_{k}^{\mathrm{BlurMix}}}\left|\right|f_{\theta^{\prime }}\left(x\right)-y{\left|\right|}_{1}$        Compute adaptive parameters $\theta^{\prime }$:        $\theta^{\prime }=\theta^{\prime }-\alpha \nabla_{\theta^{\prime }}L_{\mathrm{inner}}\left(f_{\theta^{\prime }}\right)$    **end for**    Update θ:    $\theta \leftarrow \theta -\beta \nabla_{\theta^{\prime }}L_{\mathrm{meta}}\left(f_{\theta^{\prime }}\right)$ with $L_{\mathrm{meta}}=\frac{1}{N_{GoPro}}\sum_{\left(x,y\right)\in B_{i}^{\mathrm{GoPro}}}\left|\right|f_{\theta^{\prime }}\left(x\right)-y{\left|\right|}_{1}$**end for**

The meta-tuning process begins by initializing SFSNet with pretrained weights from the GoPro dataset ($f_{\theta }$), leveraging its high deblurring performance. During training, the algorithm alternates between batches sampled from the GoPro dataset ($D_{GoPro}$) and the BlurMix dataset ($D_{BlurMix}$), simultaneously learning of domain-specific deblurring and generalization to various blur characteristics. For each GoPro batch sampled, the inner loop first clones the current model parameters $\theta^{\prime }\leftarrow \theta$ to create a task-specific copy of the SFSNet deblurring model. Subsequently, the inner loop iteratively adapts $\theta^{\prime }$ across *K* steps using batches of BlurMix ($B_{k}^{BlurMix}$). At each adaptation step, the inner loss $L_{inner}$ is computed as the mean $L_{1}$-norm reconstruction error between the batch of $N_{BlurMix}$ restored images and their corresponding ground truth sharp images from the BlurMix dataset, which is given by(8)$$L_{inner}=\;\frac{1}{N_{BlurMix}}\sum_{\left(x,y\right)\in B_{k}^{BlurMix}}\left|\right|f_{\theta^{\prime }}\left(x\right)-y{\left|\right|}_{1}\;.$$

The parameters $\theta^{\prime }$ are then computed using gradient descent with a step size α, which is represented as(9)$$\theta^{\prime }=\theta^{\prime }-\alpha \nabla_{\theta^{\prime }}L_{\mathrm{inner}}\left(f_{\theta^{\prime }}\right)\;.$$

This adaptation phase allows the model to specialize in handling diverse blur patterns present in the BlurMix dataset. After completion of the inner loop adaptation, the outer loop evaluates the performance of the adapted model ($f_{\theta^{\prime }}$) using batches from the GoPro dataset ($B_{i}^{\mathrm{GoPro}}$). The meta loss ($L_{\mathrm{meta}}$) is computed, which measures the generalization performance of the model to the GoPro domain based on the batch size $N_{\mathrm{GoPro}}$, which is given by(10)$$L_{meta}=\;\frac{1}{N_{GoPro}}\sum_{\left(x,y\right)\in B_{i}^{GoPro}}\left|\right|f_{\theta^{\prime }}\left(x\right)-y{\left|\right|}_{1}\;.$$

The meta-loss is a function of the adapted parameter $\theta^{\prime }$, which depends iteratively on the initial parameter θ. Consequently, computing the meta-gradient $\nabla_{\theta }L_{\mathrm{meta}}\left(f_{\theta^{\prime }}\right)$ requires second-order derivatives to account for the dependency of $\theta^{\prime }$ on θ, which involves computationally expensive Hessian calculations. Building on the theoretical foundations of Finn et al. [48] and Fallah et al. [49], we employ a first-order approximation (FOMAML) to reduce this computational complexity, approximating $\nabla_{\theta }L_{\mathrm{meta}}\left(f_{\theta^{\prime }}\right)$ with $\nabla \theta^{\prime }L_{\mathrm{meta}}\left(f_{\theta^{\prime }}\right)$, which simplifies the meta-update by neglecting second-order terms, assuming that the adapted parameters $\theta^{\prime }$ are sufficiently close to θ after *K* inner loop steps. This results in the following update rule:(11)$$\theta \leftarrow \theta -\beta \nabla_{\theta^{\prime }}L_{\mathrm{meta}}\left(f_{\theta^{\prime }}\right)\;.$$

This first-order approximation is both computationally efficient and effective, as it enables scalable meta-tuning of SFSNet while preserving robust generalization across diverse blur patterns, as validated by empirical results on the GoPro and BlurMix datasets.

The meta-tuning method strategy alternates between the inner and outer loops, progressively refining the adaptability of the deblurring model to various blur conditions while preserving its strong deblurring performance on the GoPro dataset. This framework effectively integrates domain-specific adaptation with the generalization benefits of meta-learning, significantly enhancing the suitability of the model for real-world scenarios characterized by various blur types. Figure 3 illustrates the alternating meta-tuning process for SFSNet across the GoPro and BlurMix datasets.

## 4. Experiments

In this section, we describe the experimental setup, including the implementation details of the SFSNet and evaluation metrics. Subsequently, a comprehensive analysis of the SFSNet with meta-tuning is provided.

### 4.1. Implementation Details and Evaluation Metrics

#### 4.1.1. Implementation Details

Following the deep-learning-based framework for image deblurring [20], we trained the SFSNet on the GoPro dataset, which consisted of 2103 pairs of blurry and sharp images. The images were cropped into 256×256 patches and randomly augmented with horizontal and vertical flips, as well as rotations. The SFSNet was trained for 300 epochs with a batch size of 4 using the Adam optimizer [50] (with decay rates $\beta_{1}=0.9$ and $\beta_{2}=0.999$). The loss function $L_{SFSNet}$ (defined in Equation Equation 5) was optimized with an initial learning rate of $2\times 10^{-4}$, which was gradually reduced to $10^{-7}$. The first layer was initialized with 48 channels, and both the encoder and decoder comprised eight blocks for the first three scales, with four blocks for the final scale.

For the meta-tuning process, we initialized the network with the pretrained SFSNet weights and then performed meta-tuning using the BlurMix dataset for adaptation and the GoPro dataset for meta-updating, as described in Algorithm 1. In this process, the inner loop updates were carried out using an initial learning rate of $\alpha =10^{-3}$ and repeated for K=2 steps, while a learning rate of $\beta =10^{-4}$ was employed for the outer updates. Meta-tuning was performed for a single epoch, processing every image from both the GoPro and BlurMix datasets exactly once with batch sizes of four and two, respectively. This method enables the SFSNet to rapidly adapt to new blur domains without full optimization for each domain. All experiments are conducted on an NVIDIA GeForce RTX 4090 GPU with the PyTorch 2.5.1 machine learning framework.

#### 4.1.2. Evaluation Metrics

The performance of our image deblurring method was evaluated quantitatively using widely adopted metrics for image deblurring, following the guidelines provided by each benchmark dataset. These metrics include the average peak signal-to-noise ratio (PSNR) and structural similarity index measure (SSIM) [51,52], which were calculated for the full-resolution restored images and their corresponding ground truth images. The PSNR is expressed in decibels (dB) and is formulated as(12)$$PSNR\left(S,\hat{I}\right)=10\log_{10}\left(\frac{255^{2}}{MSE(S,\hat{I})}\right)\;,$$
where the value 255 represents the maximum pixel intensity in an 8-bit image. The mean squared error (MSE) quantifies the average squared difference between the pixel values of the ground truth image *S* and restored image $\hat{I}$ and is expressed as(13)$$MSE\left(S,\hat{I}\right)=\frac{1}{HW}\sum_{i=1}^{H}\sum_{j=1}^{W}{\left(S_{ij}-{\hat{I}}_{ij}\right)}^{2}\;,$$
where *H* and *W* denote the height and width of the image, respectively, and $S_{ij}$ and ${\hat{I}}_{ij}$ represent the pixel intensities at positions (i,j). Lower MSE values indicate a higher similarity, which results in a higher PSNR value, signifying better restoration quality.

Whereas PSNR focuses solely on pixel-wise differences, SSIM considers structural information by comparing the luminance, contrast, and structural patterns between the ground truth and restored images. The SSIM is given by(14)$$SSIM\left(S,\hat{I}\right)=\frac{\left(2\mu_{S}\mu_{\hat{I}}+C_{1}\right)\left(2\sigma_{S\hat{I}}+C_{2}\right)}{({\mu_{S}}^{2}+\mu_{\hat{I}}^{2}+C_{1})\left({\sigma_{S}}^{2}+{\sigma_{\hat{I}}}^{2}+C_{2}\right)}\;,$$
where $\mu_{S}$ and $\mu_{\hat{I}}$ are the mean values of ground truth *S* and restoration $\hat{I}$ images, respectively. The terms $\sigma_{S}$ and $\sigma_{\hat{I}}$ denote the standard deviations, while $\sigma_{S\hat{I}}$ represents the covariance between the ground truth and restoration images. Constants $C_{1}$ and $C_{2}$ are small positive values introduced to avoid division by zero when the denominator approaches zero. The PSNR provides an objective assessment of restoration accuracy, whereas the SSIM offers insights into perceptual similarity by considering human visual preferences. Higher values for both metrics indicate that the restored image closely resembles the ground truth image in terms of numerical accuracy and visual quality. Together, these metrics offer a comprehensive evaluation of the deblurring performance.

Furthermore, the learned perceptual image patch similarity (LPIPS) metric [53] was employed to evaluate perceptual quality. LPIPS quantifies perceptual similarity between restored and ground-truth images using deep neural networks trained on human visual perception data, capturing high-level perceptual features that align closely with human judgment. Lower LPIPS scores indicate superior perceptual quality. In addition, the model size, measured in the millions of parameters (M), and computational cost, measured in gigaflops (GFLOPs), were analyzed to ensure a comprehensive assessment of the deblurring efficiency of deep-learning models.

### 4.2. Study of SFSNet Without Meta-Tuning

#### 4.2.1. Quantitative Comparison

To evaluate the effectiveness of the SFSNet, we trained the model on the GoPro dataset and compared its performance with those of state-of-the-art CNN-based image deblurring methods such as DeblurGAN-v2 [21], DBGAN [22], MTRNN [23], MIMO-UNet [24], HINet [26], BANet [27], and FSNet [29]. Comparisons were conducted on two widely used benchmark datasets, GoPro [20] and HIDE [43], both of which were generated using a frame averaging method to simulate motion-induced blur in dynamic scenes. For a fair and consistent evaluation, the results of the methods were obtained either from the original publications or generated using publicly available pretrained models released by the respective authors. This ensured that all the methods were evaluated under comparable conditions and adhered to their respective experimental protocols.

Table 5 presents a quantitative comparison between the SFSNet and other CNN-based image deblurring methods tested on the GoPro and HIDE datasets. The SFSNet achieved the highest PSNR and SSIM and the lowest LPIPS values for both datasets, demonstrating its superior deblurring performance. Specifically, on the GoPro dataset, the SFSNet attained a PSNR of 33.34 dB, an SSIM of 0.963, and an LPIPS of 0.080, outperforming the other methods. Similarly, on the HIDE dataset, the SFSNet achieved a PSNR of 31.14 dB, an SSIM of 0.941, and an LPIPS of 0.103. In addition to its high restoration quality, the SFSNet exhibited an efficient design with a parameter count of 15.10 M and a computational cost of 75 GFLOPs. The computational complexity of the SFSNet was significantly lower than that of other methods while maintaining superior deblurring performance. These results underscore the effectiveness of the SFSNet in achieving a balance between deblurring performance and computational efficiency, making it highly suitable for practical applications.

#### 4.2.2. Qualitative Comparison

Figure 4 shows a qualitative comparison of the deblurring results produced by SFSNet and several state-of-the-art methods tested on the GoPro and HIDE test images. These visual examples highlight the superior capability of the SFSNet in restoring the sharpness, texture, and fine details in challenging scenarios.

As shown in Figure 4a, the SFSNet demonstrated the capability to reconstruct the sharpness and texture of the car, which were otherwise poorly restored by other methods. Even though other methods left residual blurriness or failed to accurately recover the edges of the moving cars and the intricate patterns of the rearview mirror, the SFSNet could produce sharper and more natural results, emphasizing its capacity for precise deblurring.

Similarly, as shown in Figure 4b, the SFSNet excelled in restoring facial details and object textures. Our method effectively reconstructed the facial details of the woman and the fine details of the coat texture, both of which were only partially recovered or missed by other models. This indicates the ability of the SFSNet to handle complex blur patterns and enhance fine-grained details while preserving naturalistic image quality.

#### 4.2.3. Ablation Study of the Spatial Gated Module (SGM) in the Regional Feature Extractor (RFE) Module

We conducted ablation studies to analyze the effects of different design components on the deblurring performance of the SFSNet, and the results are tabulated in Table 6. In the ablation study, we replaced the Spatial Gated Module (SGM) with a 3×3 depthwise (DW) convolution or multirate dilated convolutions in the Region Feature Extraction (RFE) module. The results were evaluated in terms of the PSNR, SSIM, parameter count, and computational complexity (GFLOPs), where models were trained and tested on the GoPro dataset.

The first ablation, where we used a standard 3×3 DW convolution for feature extraction, achieved a PSNR of 33.22 dB and an SSIM of 0.962 with a minimal number of parameters (14.92 M) and computational complexity (74.57 GFLOPs). Although this configuration maintained low resource usage, its deblurring performance was slightly inferior to that of the proposed SGM.

The second ablation, where we employed multirate dilated convolutions with a rate of 1,2,3, and 4, resulted in a marginally improved PSNR of 33.24 dB and an SSIM of 0.962. Even though this approach expanded the receptive field, it significantly increased the parameter count (45.64 M) and computational complexity (180.29 GFLOPs), making it computationally expensive and less efficient than other configurations.

In contrast, the proposed SGM achieved the highest PSNR (33.34 dB) and SSIM (0.964), demonstrating its effectiveness in capturing richer spatial contexts and selecting relevant features for deblurring. Despite its slightly higher parameter count (15.10 M) compared with that of the 3×3 DW convolution, its computational cost (75.10 GFLOPs) was comparable to that of a lighter configuration.

### 4.3. Study of SFSNet with Meta-Tuning

#### 4.3.1. Quantitative Comparison

Deep-learning-based methods trained on the GoPro dataset often exhibit strong performance within a specific blur domain. However, these methods tend to overfit the GoPro dataset and fail to generalize effectively to other blur domains, particularly in real-world scenarios. This limitation frequently results in undesired artifacts and poor deblurring performance when applied to diverse or complex blur patterns. To address this issue, we applied meta-tuning to the SFSNet pretrained on the GoPro dataset to enhance its adaptability across various blur domains. Meta-tuning requires only a single epoch for tuning, which makes it computationally efficient while delivering significant improvements in performance.

The integration of meta-tuning into the SFSNet significantly enhanced its ability to adapt to diverse blur domains. This transformative impact is evident from the quantitative comparison results presented in Table 7, which were evaluated using the REDS [44], RealBlur [42], RSBlur [45], and ReLoSBlur [46] testing sets.

Meta-tuning enabled the SFSNet to dynamically adjust its parameters through a meta-learning framework, which enhanced its performance across synthetic and real-world datasets. On the REDS dataset, which is a synthetic motion blur benchmark, the MetaSFSNet* achieved substantial improvements of +2.97 dB in PSNR and +0.034 in SSIM compared with the baseline SFSNet. This improvement demonstrates enhanced generalization of the model in structured motion blur scenarios. Similarly, on the RealBlur dataset, which contains complex real-world blurs, the MetaSFSNet* achieved significant gains of +1.56 dB in PSNR and +0.027 in SSIM. These results highlight its ability to adapt to nonuniform and natural blur variations. Moreover, the improvements on the RSBlur (PSNR: +3.12 dB, SSIM: +0.095) and ReLoBlur (PSNR: +2.87 dB, SSIM: +0.051) datasets further proved the effectiveness of meta-tuning in addressing diverse and challenging blur types. In addition, these adaptation results were achieved within only one epoch of meta-tuning on the new blur domains, rather than through a full optimization process aimed at obtaining the best possible results for each domain. This rapid adaptation, although not exhaustive, demonstrates the efficiency and practicality of the meta-tuning strategy for real-world applications, where swift adjustment to novel blur distributions is often critical. The consistent performance improvements across the datasets indicate that meta-tuning endows the network with a robust mechanism for domain adaptation, allowing it to retain valuable information from pretraining while quickly adapting to new, unseen blur characteristics.

#### 4.3.2. Qualitative Comparison

Figure 5 demonstrates the effectiveness of applying meta-tuning to the pretrained SFSNet, resulting in the enhanced MetaSFSNet* model. On the REDS dataset, deep learning methods trained solely on the GoPro dataset struggled to deblur images affected by motion blurring, as shown in Figure 5a. In contrast, the MetaSFSNet* successfully restored both the texture and clarity of the scene objects. In real-world datasets, models trained on the GoPro dataset often generate undesired artifacts when unknown blur characteristics are encountered. For example, when tested on the RSBlur dataset as shown in Figure 5b, methods such as DBGAN, MIMO-UNet+, HINet, and BANet+ failed to handle complex blurs and introduced additional artifacts. Similarly, nearly all of the methods produced localized artifacts in the ReLoBlur dataset, as shown in Figure 5c. When tested on the RealBlur dataset, DBGAN, BANet+, FSNet, and SFSNet models exhibited limitations and introduced regional artifacts, as shown in Figure 5d. Notably, when tested on the RWBI dataset, which included images collected from real handheld devices, all deep-learning-based methods trained solely on the GoPro dataset failed to recover the sharpness of objects and details in the image panels. However, with meta-tuning, the MetaSFSNet* addressed these limitations and effectively restored the fine details, demonstrating its superior adaptability to diverse and challenging blur domains with a single epoch of tuning. These results underscore the critical role of meta-tuning in enhancing the performance of the SFSNet across various datasets and real-world scenarios.

#### 4.3.3. Comparison Between Meta-Tuning and Fine-Tuning Methods

Comprehensive ablation studies were conducted by comparing the different tuning approaches to investigate the effectiveness of our proposed meta-tuning strategy. Table 8 presents a quantitative comparison between fine-tuning and our meta-tuning strategy, MetaSFSNet*, across two categories of deblurring datasets: pretrained blur domains (GoPro) and new blur-adaptive domains (REDS, RealBlur, RSBlur, and ReLoBlur). Both methods were initialized with identical, pretrained weights from SFSNet, trained for 300 epochs on the GoPro dataset. For the experiments, fine-tuning employed a learning rate of $\beta =10^{-3}$ for one and three epochs on the BlurMix dataset, as well as on a combination of BlurMix and GoPro, matching the learning rate used in the inner loop of the meta-tuning strategy.

The ablation studies provide critical insights into the performance of fine-tuning and meta-tuning strategies. The pretrained SFSNet baseline excels on the GoPro dataset (PSNR: 33.34, SSIM: 0.963, LPIPS: 0.081) but struggles to generalize to new blur-adaptive domains, yielding lower metrics (e.g., PSNR: 26.91 on REDS, 28.04 on RSBlur). Fine-tuning SFSNet on the BlurMix dataset (FineSFSNet†) improves adaptation to new domains, yet reveals trade-offs. For example, one epoch of fine-tuning produces marginal gains (e.g., PSNR: 27.09 on REDS), whereas three epochs enhance new-domain performance (e.g., PSNR: 31.02 on RSBlur) but degrade GoPro results (PSNR drop: 1.65), indicating catastrophic forgetting. Fine-tuning SFSNet on both GoPro and BlurMix datasets (FineSFSNet*) reduces this degradation but substantially increases training time (e.g., 1006 s for three epochs).

In contrast, MetaSFSNet* achieves a superior balance between retaining pretrained domain performance and adapting to new blur types. After one epoch, MetaSFSNet* outperforms the baseline on new domains, with gains of up to 3.12 dB in PSNR on RSBlur (31.16 dB vs. 28.04 dB) and 2.87 dB in PSNR on ReLoBlur (32.32 dB vs. 29.45 dB), while experiencing only a modest decline on GoPro (PSNR: 32.61 dB, a 0.73 dB drop). Meta-tuning for three epochs (MetaSFSNet-3*) further improves performance across all new domains, securing the highest scores on REDS (PSNR: 30.44 dB, SSIM: 0.919, LPIPS: 0.132), RealBlur (PSNR: 30.66 dB, SSIM: 0.905, LPIPS: 0.133), RSBlur (PSNR: 31.29 dB, SSIM: 0.824, LPIPS: 0.275), and ReLoBlur (PSNR: 33.05 dB, SSIM: 0.921, LPIPS: 0.143), while maintaining stable GoPro performance (PSNR: 32.65 dB).

Moreover, MetaSFSNet* demonstrates remarkable computational efficiency. With a training time of 401 s for one epoch, MetaSFSNet* surpasses the fine-tuning model for three epochs (1006 s) across most new domains. Additionally, SSIM and LPIPS metrics align with PSNR trends, confirming that meta-tuning enhances not only pixel-wise accuracy but also structural similarity and perceptual quality. The performance gap between MetaSFSNet* and SFSNet with fine-tuning widens on datasets with diverse blur patterns, suggesting that meta-tuning adapts more effectively to varied blur characteristics.

#### 4.3.4. Hyperparameter Selection for MetaSFSNet*

To optimize the performance of MetaSFSNet*, we assess the impact of key hyperparameters: batch size, number of inner steps, and inner-loop learning rate (α). The batch size and inner steps determine the number of image pairs from the BlurMix dataset used in meta-tuning, while α governs the update rate in the inner loop. For MetaSFSNet*, we set the outer loop learning rate to 1e−4, the outer loop batch size to 4, and configure the inner loop with a batch size of 2 and two inner steps to balance the number of image pairs tuned on the GoPro and BlurMix datasets in each loop. Specifically, one inner step processes half as many BlurMix image pairs as GoPro in the outer loop, whereas two inner steps align the number of BlurMix pairs with the outer-loop GoPro pairs. Table 9 presents an ablation study that compares the configurations of the inner steps (1 or 2) and α (ranging from 1e−2 to 1e−4) on the GoPro [20] and RealBlur [42] datasets, evaluated using PSNR, SSIM, and LPIPS metrics, along with the runtime in seconds.

The ablation study yields critical insights into hyperparameter optimization for MetaSFSNet*. Configurations with two inner steps consistently outperform those with one inner step across both datasets, achieving higher PSNR and SSIM and lower LPIPS values. This improvement stems from the increased number of BlurMix image pairs processed, which aligns the tuning volume with the GoPro outer loop, thereby enhancing adaptation to new blur domains. Runtime analysis reveals that one inner step requires less computational time (324–326 s) than two inner steps (401–418 s) due to processing fewer image pairs.

Based on these results, we select two inner steps with α = 1e−3 (401 s) for MetaSFSNet*, as it achieves the best SSIM and LPIPS on RealBlur while maintaining strong GoPro performance and computational efficiency. This configuration optimizes cross-domain deblurring by effectively balancing adaptation to new blur domains with retention of pretrained domain performance.

## 5. Conclusions

CNN-based methods have demonstrated remarkable success in image deblurring. However, their reliance on small kernel sizes inherently limits the receptive field, restricting their ability to effectively capture the contextual information necessary for high-quality deblurring. Moreover, deep-learning-based approaches, often trained on synthetic datasets, exhibit significant overfitting, leading to poor performance and artifact generation when applied to diverse real-world blur domains. In this paper, we introduced the Spatial Feature Selection Network (SFSNet), which enhances deblurring performance by incorporating a Regional Feature Extractor (RFE) module. The RFE expands the receptive field and adaptively selects critical spatial features to address the limitations of conventional CNNs. In addition, we introduced a meta-tuning strategy for the SFSNet by leveraging the BlurMix dataset to enable effective adaptation to various blur types. This approach minimizes the training effort required for new datasets while improving the robustness and generalizability of the model, thereby enabling it to handle the complexities of diverse blur characteristics in real-world scenarios. 

## Figures and Tables

**Figure 1 sensors-25-05211-f001:**
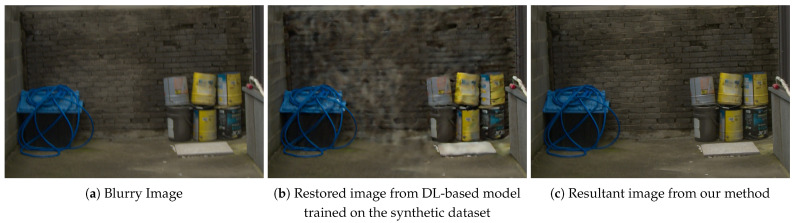
Deblurred results on real-world blur image. (**a**) Input blurry image. (**b**) Image restored by DL-based image deblurring methods trained on the synthetic dataset may produce artifacts and degraded performance on unseen blur domains. (**c**) Images restored by our pretrained model, after meta-tuning in a single epoch, are free of artifacts and show improved performance.

**Figure 2 sensors-25-05211-f002:**
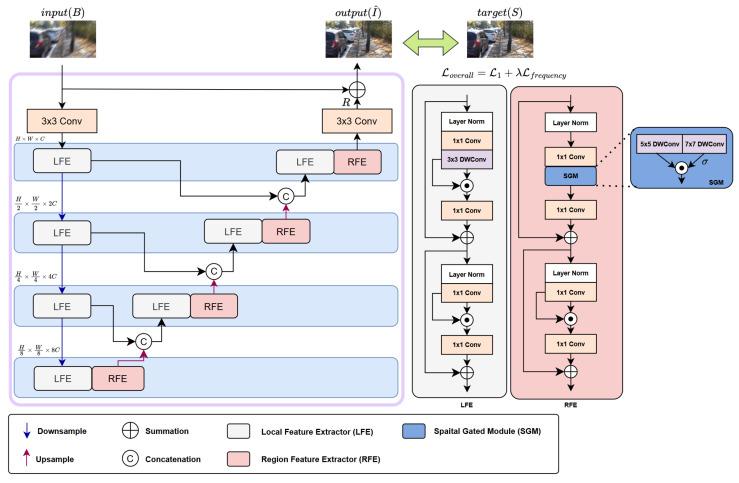
Detailed architecture of the proposed SFSNet. The network adopts a multi-scale encoder–decoder structure, incorporating local feature extractors (LFE) and regional feature extractors (RFE) at multiple hierarchical levels. The LFE modules capture fine-grained local details, while the RFE modules extract and emphasize broader contextual features.

**Figure 3 sensors-25-05211-f003:**
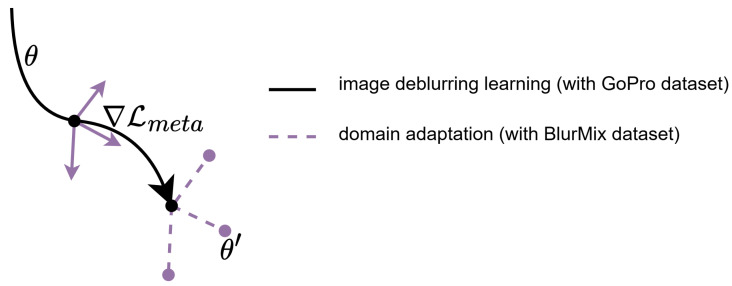
Overview of the meta-tuning strategy for SFSNet. The process alternates between inner loop adaptation on the BlurMix dataset, which optimizes the model for diverse blur patterns, and outer loop meta-updates on the GoPro dataset, which ensure robust generalization to high-quality deblurring. This alternating optimization enables SFSNet to balance domain-specific adaptation with broad applicability across varied blur characteristics.

**Figure 4 sensors-25-05211-f004:**
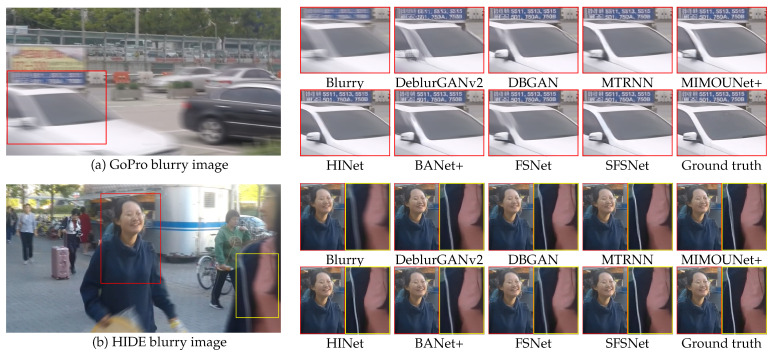
Qualitative comparison of the DL-based image deblurring models tested on the GoPro and HIDE test images.

**Figure 5 sensors-25-05211-f005:**
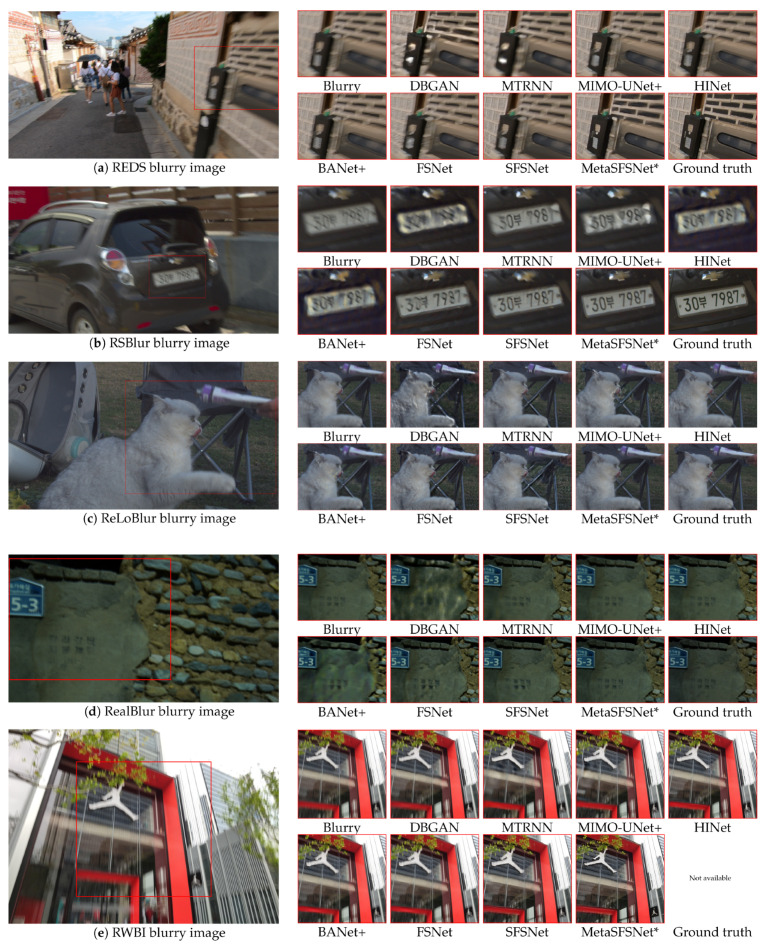
Qualitative comparison of image deblurring models tested on the REDS, RSBlur, ReLoBlur, RealBlur, and RWBI test images. The MetaSFSNet* indicates that the SFSNet includes meta-tuning.

**Table 1 sensors-25-05211-t001:** Properties and acquisition methods of existing motion blur datasets. The symbol ‘**✗**’ indicates datasets lacking available training or testing sets, while ‘**✓**’ denotes datasets with available training and testing sets.

Dataset	Splitting of Training/Testing Sets	Blur Model	Acquisition Method
Levin [8]	**✗**	Uniform	Kernel convolution
Sun [11]	**✗**	Uniform	Kernel convolution
Kohler [12]	**✗**	Nonuniform	Kernel convolution
MC-UHDM [14]	**✓**	Nonuniform	Kernel convolution
GoPro [20]	**✓**	Nonuniform	Frame averaging
HIDE [43]	**✓**	Nonuniform	Frame averaging
REDS [44]	**✓**	Nonuniform	Frame averaging
RealBlur [42]	**✓**	Nonuniform	Real-world shooting
RealBlur-Tele [42]	**✗**	Nonuniform	Real-world shooting
RSBlur [45]	**✓**	Nonuniform	Real-world shooting
ReLoBlur [46]	**✓**	Nonuniform	Real-world shooting
RWBI [22]	**✗**	Nonuniform	Real-world shooting

**Table 2 sensors-25-05211-t002:** Performance comparison of deep learning models trained on different types of datasets.

Model	Training Dataset	Testing Dataset	PSNR↑	SSIM↑
MIMO-UNet+ [24]	GoPro [20]	RealBlur [42]	27.63	0.837
	RealBlur [42]	RealBlur [42]	31.92	0.919
BANet+ [27]	GoPro [20]	RealBlur [42]	28.10	0.852
	RealBlur [42]	RealBlur [42]	32.40	0.929
RealBlur testing set [42]	-	RealBlur [42]	27.68	0.825
MIMO-UNet+ [24]	GoPro [20]	RSBlur [45]	26.92	0.721
	RSBlur [45]	RSBlur [45]	33.37	0.856
FSNet [29]	GoPro [20]	RSBlur [45]	28.21	0.728
	RSBlur [45]	RSBlur [45]	34.31	0.872
RSBlur testing set [45]	-	RSBlur [45]	29.03	0.739

**Table 3 sensors-25-05211-t003:** Datasets used in this paper. The asterisk (*) denotes the dataset collected in the real world without ground truth images, used for qualitative evaluation.

Dataset	Training/Testing Image Pairs	Resolution	Acquisition Method
MC-UHDM [14]	8000/2000	3984 × 2656	Kernel convolution
GoPro [20]	2103/1111	1280 × 720	Frame averaging
HIDE [43]	6397/2025	1280 × 720	Frame averaging
REDS [44]	24,000/3000	1280 × 720	Frame averaging
RealBlur [42]	3758/980	680 × 773	Real-world shooting (wide-angle lens)
RealBlur-Tele [42]	996	950 × 598	Real-world shooting (telephoto lens)
RSBlur [45]	8878/3360	1920 × 1200	Real-world shooting
ReLoBlur [46]	2010/395	2152 × 1436	Real-world shooting (local blurs)
RWBI * [22]	3112	1000 × 750	Real-world shooting (handheld devices)

**Table 4 sensors-25-05211-t004:** Number of image pairs, resolutions, and acquisition methods used to construct the BlurMix dataset.

Dataset	Number of Image Pairs in the Training Set	Resolution	Acquisition Method
MC-UHDM [14]	403	3984×2656	Kernel convolution
REDS [44]	500	1280×720	Frame averaging
RealBlur [42]	400	680×773	Real-world shooting (wide-angle lens)
RealBlur-Tele [42]	400	950×598	Real-world shooting (telephoto lens)
RSBlur [45]	400	1920×1200	Real-world shooting
Total image pairs in the BlurMix dataset:	2103		

**Table 5 sensors-25-05211-t005:** Quantitative comparison of deep-learning-based image deblurring methods tested on the GoPro and HIDE testing sets. The **bold** values indicate the best results. The symbol “↑” signifies that higher values are better, while “↓” indicates that lower values are preferable.

Methods	GoPro [20]			HIDE [43]			#Params↓ (M)	Complexity↓ (GFLOPs)
	PSNR↑	SSIM↑	LPIPS↓	PSNR↑	>SSIM↑	LPIPS↓		
DeblurGAN-v2 [21]	29.55	0.934	0.117	26.61	0.875	0.158	60.9	411
DBGAN [22]	31.10	0.942	0.109	28.94	0.915	0.142	11.6	760
MTRNN [23]	31.15	0.945	0.122	29.15	0.918	0.154	**2.64**	164
MIMO-UNet+ [24]	32.45	0.958	0.090	29.99	0.930	0.124	16.1	145
HINet [26]	32.71	0.959	0.088	30.32	0.932	0.119	88.7	171
BANet+ [27]	33.03	0.961	0.085	30.58	0.935	0.117	40.0	588
FSNet [29]	33.28	**0.963**	**0.080**	31.05	**0.941**	0.109	13.28	111
SFSNet (ours)	**33.34**	**0.963**	**0.080**	**31.14**	**0.941**	**0.103**	15.10	**75**

**Table 6 sensors-25-05211-t006:** Effects of ablation on the deblurring performance of the SFSNet.

Ablation	GoPro [20]		#Params↓ (M)	Complexity↓ (GFLOPs)
	PSNR↑	SSIM↑		
With 3×3 DW convolutions	33.22	0.962	**14.92**	**74.57**
With multirate (1, 2, 3, 4) dilated convolutions [33]	33.24	0.962	45.64	180.29
With SGM (ours)	**33.34**	**0.964**	15.10	75.10

**Table 7 sensors-25-05211-t007:** Quantitative comparison of the deblurring performance of image deblurring methods tested on different datasets. The red values indicate the different deblurring performance as a result of incorporating meta-tuning in a meta epoch of the image deblurring methods. The **bold** values indicate the best results. The asterisk (*) indicates that meta-tuning was applied, alternating between the inner and outer loops, on the GoPro and BlurMix datasets.

Methods	REDS [44]		RealBlur [42]		RSBlur [45]		ReLoBlur [46]	
	PSNR↑	SSIM↑	PSNR↑	SSIM↑	PSNR↑	SSIM↑	PSNR↑	SSIM↑
DBGAN [22]	22.95	0.788	24.93	0.745	27.15	0.709	23.64	0.825
MTRNN [23]	26.83	0.864	28.44	0.862	28.79	0.741	31.24	0.900
MIMO-UNet+ [24]	26.43	0.859	27.63	0.837	26.92	0.721	28.14	0.864
HINet [26]	26.77	0.867	28.17	0.849	28.82	0.716	32.19	0.894
BANet+ [27]	27.01	0.873	28.10	0.852	26.34	0.695	28.34	0.827
FSNet [29]	26.70	0.869	28.47	0.868	28.21	0.728	28.80	0.862
SFSNet (ours)	26.91	0.878	28.64	0.873	28.04	0.721	29.45	0.866
MetaSFSNet *	**29.88**	**0.912**	**30.20**	**0.900**	**31.16**	**0.816**	**32.32**	**0.917**
(our method with meta-tuning)	(+2.97)	(+0.034)	(+1.56)	(+0.027)	(+3.12)	(+0.095)	(+2.87)	(+0.051)

**Table 8 sensors-25-05211-t008:** Quantitative comparison of the deblurring performance of the FineSFSNet (SFSNet with fine-tuning) and MetaSFSNet (SFSNet with meta-tuning) tested on various datasets. The **bold** values indicate the best results. The underline denotes the second-best results. The dagger (†) indicates that tuning was applied only to the BlurMix dataset. The asterisk (*) denotes that tuning was applied to both the GoPro and BlurMix datasets.

Ablations	Pre-Trained Blur Domain			New Blur Adaptive Domains												Runtime (Seconds)
	GoPro [20]			REDS [44]			RealBlur [42]			RSBlur [45]			ReLoBlur [46]			
	PSNR↑	SSIM↑	LPIPS↓	PSNR↑	SSIM↑	LPIPS↓	PSNR↑	SSIM↑	LPIPS↓	PSNR↑	SSIM↑	LPIPS↓	PSNR↑	SSIM↑	LPIPS↓	
Pretrained SFSNet (baseline)	**33.34**	**0.963**	**0.081**	26.91	0.878	0.185	28.64	0.873	0.159	28.04	0.721	0.317	29.45	0.866	0.250	-
FineSFSNet † (1 epoch)	33.33	**0.963**	0.083	27.09	0.879	0.187	28.71	0.875	0.158	29.05	0.741	0.284	30.15	0.882	0.224	183
FineSFSNet † (3 epochs)	31.69	0.954	0.123	29.88	0.910	0.150	30.19	0.897	0.150	31.02	0.815	0.311	30.80	0.915	0.275	526
FineSFSNet * (1 epoch)	32.84	0.960	0.088	28.97	0.901	0.167	29.63	0.890	0.157	30.39	0.798	0.306	31.89	0.915	0.245	339
FineSFSNet * (3 epoch)	32.90	0.961	0.086	29.57	0.909	0.155	30.00	0.894	0.152	30.90	0.811	0.315	32.28	0.917	0.238	1006
MetaSFSNet * (1 epoch)	32.61	0.959	0.085	29.88	0.912	0.144	30.20	0.900	0.136	31.16	0.816	**0.268**	32.32	0.917	0.173	401
MetaSFSNet-3 * (3 epoch)	32.65	0.959	0.086	**30.44**	**0.919**	**0.132**	**30.66**	**0.905**	**0.133**	**31.29**	**0.824**	0.275	**33.05**	**0.921**	**0.143**	1206

**Table 9 sensors-25-05211-t009:** Ablation study of hyperparameter configurations for MetaSFSNet* on GoPro and RealBlur datasets. **Bold** values denote the best results.

Inner Steps	α	GoPro [20]			RealBlur [42]			Runtime (Seconds)
		PSNR↑	SSIM↑	LPIPS↓	PSNR↑	SSIM↑	LPIPS↓	
1	1e−2	32.47	0.958	0.088	30.10	0.897	0.143	326
1	1e−3	32.49	**0.959**	0.087	30.09	0.895	0.143	325
1	1e−4	32.45	0.958	0.087	29.84	0.892	0.144	324
2	1e−2	**32.63**	**0.959**	0.086	30.11	0.898	0.138	418
2	1e−3	32.61	**0.959**	0.085	30.20	**0.916**	**0.136**	401
2	1e−4	32.62	**0.959**	**0.084**	**30.22**	0.900	0.137	403

## Data Availability

Data are contained within the article.

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
