# Peer review of "Adaptive Image Deblurring Convolutional Neural Network with Meta-Tuning"

_sensors, 2025, doi:10.3390/s25165211_

Round 1
Reviewer 1 Report
Comments and Suggestions for Authors
- The author state that "deep learning-based approaches, often trained on synthetic datasets, exhibit significant overfitting, leading to poor performance and artifact generation when applied to diverse real-world blur domains". From the perspective of algorithm design, I do not believe that this paper has solved the problem. Please provide a more detailed explanation.
- In Page 7, we categorized existing deblurring datasets based on their acquisition methods and selected nonuniform datasets that are suitable for training and testing deep learning-based models for image deblurring. Will this approach affect the final verification results?
- The total amount of dataset used in this paper is not large. Does this indicate the effectiveness of the algorithm, while ensuring the fairness of the comparison results.
- The correctness of 11 needs to be discussed.
Reviewer 2 Report
Comments and Suggestions for Authors
The manuscript titled "Adaptive Image Deblurring Convolutional Neural Network with Meta-Tuning" addresses an important challenge in image processing, which is improving motion deblurring performance and model generalization across various blur domains. The authors propose the Spatial Feature Selection Network (SFSNet) enhanced by a Regional Feature Extractor (RFE) to expand the receptive field and selectively emphasize spatially important features.
Strengths of the paper include a well-motivated problem statement, innovative combination of receptive field expansion and meta-learning strategies, and comprehensive experimental results demonstrating substantial reductions in artifacts and improved deblurring across multiple datasets. The proposed meta-tuning approach is especially promising for practical applications requiring rapid model adaptation.
However, the manuscript can be improved in several aspects. The introduction could better situate the work within the broader state-of-the-art, with more emphasis on recent transformer-based and attention mechanism methods in deblurring tasks. Method descriptions, particularly of the RFE module and meta-tuning algorithm, sometimes lack sufficient detail, which hinders reproducibility – this hast to be improved. The paper would benefit from a clearer explanation of certain implementation choices and more discussion on the computational cost of the meta-tuning approach.
Reviewer 3 Report
Comments and Suggestions for Authors
This research introduces SFSNet, a novel CNN-based architecture incorporating a Spatial Feature Selection Network and Regional Feature Extractor to overcome the limitations of small receptive fields in traditional image deblurring methods. It also proposes a meta-tuning strategy using a newly constructed BlurMix dataset, enabling rapid adaptation to diverse blur domains and significantly improving generalization and performance on real-world blur scenarios.
Here are my comments:
The meta-tuning algorithm essentially approximates a MAML-like approach but lacks theoretical rigor or convergence analysis. There is no quantitative evaluation of domain shift, which weakens the claims about domain generalization. The BlurMix dataset construction lacks clear statistical analysis, making it hard to assess diversity rigorously.
The computational overhead of meta-tuning is stated to be low, but runtime benchmarks or memory footprints are not reported. No user study or perceptual evaluation such as LPIPS is provided, though SSIM and PSNR can fail to capture visual quality and naturalness. Moreover, meta-tuning vs. fine-tuning is compared only in terms of accuracy but not training time or energy efficiency, which would support practical claims.
Several redundant phrases appear in the abstract and introduction. For example, repeating the issue of small receptive fields and overfitting in multiple places. Some figures (Fig. 2 and 3) are referenced without enough description of what new insights they bring. Minor English grammar issues persist, like “debluirring” instead of “deblurring,” or inconsistent pluralization.
This paper lacks reviewing recent studies in introdution, like Predicting flow status of a flexible rectifier using cognitive computing.
The proposed meta-tuning strategy lacks mathematical novelty; it is an application of gradient-based meta-learning like MAML, yet no justification is provided for hyperparameter choices or step sizes beyond empirical tuning.
The SGM module is empirically effective, but there is no theoretical justification or intuition for why the combination of 5×5 and 7×7 depthwise convolutions performs better than other receptive field aggregation methods.
Round 2
Reviewer 2 Report
Comments and Suggestions for Authors
The authors have significantly improved the manuscript and implemented all my remarks from the first-round review. Therefore, I recommend accepting the paper.
Reviewer 3 Report
Comments and Suggestions for Authors
ok